# Identification of Genomic Signatures for Colorectal Cancer Survival Using Exploratory Data Mining

**DOI:** 10.3390/ijms25063220

**Published:** 2024-03-12

**Authors:** Justin J. Hummel, Danlu Liu, Erin Tallon, John Snyder, Wesley Warren, Chi-Ren Shyu, Jonathan Mitchem, Rene Cortese

**Affiliations:** 1Institute for Data Science and Informatics, University of Missouri, Columbia, MO 65212, USA; jjh42v@mail.missouri.edu (J.J.H.); erin.tallon@mail.missouri.edu (E.T.); warrenwc@missouri.edu (W.W.); shyuc@missouri.edu (C.-R.S.); jonathan.mitchem@va.gov (J.M.); 2Department of Electrical Engineering and Computer Science, University of Missouri, Columbia, MO 65212, USA; liudanlu001@gmail.com; 3Department of Statistics, University of Missouri, Columbia, MO 65212, USA; johnsnyder@missouri.edu; 4Division of Animal Sciences, University of Missouri, Columbia, MO 65212, USA; 5Department of Surgery, University of Missouri, Columbia, MO 65212, USA; 6Harry S. Truman Memorial Veterans’ Hospital, Columbia, MO 65212, USA; 7Siteman Cancer Center, Washington University School of Medicine, St. Louis, MO 63130, USA; 8Department of Pediatrics, University of Missouri, Columbia, MO 65212, USA; 9Department of Gynecology, Obstetrics and Women’s Health, University of Missouri, Columbia, MO 65212, USA; 10Ellis Fischel Cancer Center, University of Missouri, Columbia, MO 65212, USA

**Keywords:** colorectal cancer, prognosis, F1CDx repurposing, explainable artificial intelligence, genomic signature

## Abstract

Clinicopathological presentations are critical for establishing a postoperative treatment regimen in Colorectal Cancer (CRC), although the prognostic value is low in Stage 2 CRC. We implemented a novel exploratory algorithm based on artificial intelligence (explainable artificial intelligence, XAI) that integrates mutational and clinical features to identify genomic signatures by repurposing the FoundationOne Companion Diagnostic (F1CDx) assay. The training data set (*n* = 378) consisted of subjects with recurrent and non-recurrent Stage 2 or 3 CRC retrieved from TCGA. Genomic signatures were built for identifying subgroups in Stage 2 and 3 CRC patients according to recurrence using genomic parameters and further associations with the clinical presentation. The summarization of the top-performing genomic signatures resulted in a 32-gene genomic signature that could predict tumor recurrence in CRC Stage 2 patients with high precision. The genomic signature was further validated using an independent dataset (*n* = 149), resulting in high-precision prognosis (AUC: 0.952; PPV = 0.974; NPV = 0.923). We anticipate that our genomic signatures and NCCN guidelines will improve recurrence predictions in CRC molecular stratification.

## 1. Introduction

Colorectal cancer (CRC) is responsible for approximately 900,000 deaths around the world yearly, making CRC the second leading cause of death related to cancer [1]. Remarkably, patients with Stage 2 CRC have a different response to adjuvant chemotherapy [1,2,3,4] and a different disease-free survival [2,4,5,6,7] to those with Stage 3 cancer. Therapeutic indications, such as adjuvant systemic therapy (AST), have traditionally relied heavily upon the concrete facts of clinicopathological presentations to structure the appropriate course of treatment for patients [3]. However, the selection of the treatment in CRC can be subjective, especially for Stage 2 patients, due to the lack of tools able to predict which patients with early-stage cancers will benefit from the AST treatment [5,6,8].

New technologies harnessing high-throughput molecular diagnostics have begun to entertain alternative strategies to augment their patient selection process concerning chemotherapy treatment regimens [9,10]. The FoundationOne Companion Diagnostic (F1CDx, Roche) is a comprehensive genomic profiling platform, holding FDA-CMS approval, that can evaluate solid tumors for the benefit of additional targeted therapy for patients already receiving AST [7,11,12,13,14]. The feasibility of repositioning this diagnostic platform for calculating the probability of recurrence in CRC Stages 2 and/or 3 patients has yet to be addressed. The extension of F1CDx towards the predictive capacities of inferring cancer recurrence may improve treatment selection and prognostic assessments. However, previous studies repurposing various companion diagnostics, including F1CDx, have had limited success [11,15,16,17], suggesting that more robust computational methods are required for the building of genomic signatures for patient stratification.

Exploratory data mining algorithms such as the one developed by iDAS lab (i.e., explainable artificial intelligence, XAI [18]) enable the collection of specific groups of patients (i.e., subgroups) from different cohorts. The algorithm allows the interpretation of machine learning analysis by employing a deep exploratory mining process that creates and prioritizes potential subpopulations based on their explainable contrast patterns. According to the analytical retentions from each observed subgroup outcome, the relational elements consistent amongst both the clinical presentations and genetic mutations across each subgroup can be assessed [18]. Pattern mining enables the discovery of the non-intuitive relationships potentially able to uncover the missing link connecting cancer progression, treatment resistance, and recurrence [3,15,19,20,21]. In this study, we aimed to evaluate the translational capabilities of repurposing F1CDx and how it might examine both the phenotypic and genomic aspects of patients to better assess their risk of recurrence, particularly in CRC Stage 2 patients. To this end, we applied our novel XAI algorithm to assess F1CDx test results to build a genomic signature for determining the risk of recurrence in CRC stage 2 patients.

## 2. Results

### 2.1. Unsupervised Data Mining Algorithm Enables the Stratification of CRC Patients

Figure 1 presents an overview of the study design and analytical workflow. Our initial dataset consisted of the F1CDx marker panel (*n* = 324 genes, Appendix A) corresponding to Stage 2 (*n* = 212) and 3 (*n* = 166) CRC patients. Each CRC stage included patients with recurrent (*n* = 61 and *n* = 51 for Stage 2 and 3, respectively) and non-recurrent (*n* = 151 and *n* = 115 for Stage 2 and 3, respectively) cancers. The patient characteristics are provided in Appendix A. Mutations in each panel were combined with the most relevant clinicopathological variables (Figure 1A). Panels were implemented with the use of our unsupervised data mining algorithm (XAI) [18] to identify novel patient groups according to stage and recurrence. We discovered a sizable collection of subgroups of patients with Stage 2 and 3 CRC based on the pattern mining algorithm, which identified the best-performing genomic signatures in each CRC population subgroup (i.e., CRC Stage 2, CRC Stage 3 and CRC Stages 2 and 3 combined) (Figure 1B).

Appendix A contains the top subgroup for each of the three groups used in this study. The CRC stage 2 primary population subgroup was identified as MG02PS04. The class 01 genomic signature was defined by 56 gene mutations and class 02 comprised 30 gene mutations that were absent of all mutational elements. The CRC stage 3 primary population subgroup was identified as MG03PS19. Class 01 was defined by 11 gene mutations and class 02 comprised 5 gene mutations absent of all mutational elements. For CRC stages 2 and 3 combined, the primary population subgroup was identified as MG23PS19. Class 01 was defined by 61 gene mutations and class 02 comprised 33 gene mutations that were absent of all mutational elements. The gene composition of each genomic signature is provided in Appendix A.

### 2.2. Assessment of Predictive Capacity for CRC Staging

Figure 2 depicts the performance of the top genomic signatures for classifying recurrent and disease-free CRC patients in each subgroup in the training cohort (i.e., TCGA cohort). For each genomic signature, the Receiver Operating Curve (ROC) curve analysis was conducted using three different methods (i.e., empiric, binomial, and nonparametric) and the Area Under the Curve (AUC) was calculated. The genomic signature MG02PS04 (Figure 2A) showed high accuracy in distinguishing between recurrent and disease-free CRC Stage 2 patients (AUC = 0.9551, 0.9694, and 0.9466, for empiric, binomial, and nonparametric ROCs, respectively). In turn, genomic signatures MG03PS19 (Figure 2B; AUC = 0.8899, 0.9169, and 0.8723) and MG23PS19 (Figure 2C; AUC = 0.8488, 0.8725, and 0.8369) displayed high accuracy in distinguishing between recurrent and disease-free CRC Stage 3 patients and CRC Stage 2 and 3 patients combined, respectively. Remarkably, the genomic signature produced from the CRC Stage 2 subgroup outperformed those from the other CRC subgroups (i.e., Stage 3, and Stage 2 and 3 combined subgroups), and may have potential clinical applications for supplementing diagnostic protocols for CRC Stage 2 patients.

### 2.3. Disease-Free Survival-Based-Predictions by CRC Stage

Appendix A details the genomic signatures used for the disease-free survival analysis of each of the three CRC groups, focusing on their putative clinical indication. The gene composition of each genomic signature is provided in Appendix A.

For Stage 2 subjects, the MG02PS03 and MG02PS04 mutation-negative genomic signatures were associated with the occurrence of primary tumor and recurrence events in non-CMS1 subtype subjects, respectively, whereas mutation-positive MG02PS05 was associated with disease-free survival. For CRC Stage 3, we identified two genomic signatures with potential clinical indications: the mutation-positive MG03PS18 genomic signature was associated with 30–60 months of disease-free survival and recurrence events regardless of the CMS subtype, while the mutation-positive MG03PS18 genomic signature was associated with recurrence events in CMS1 subjects. In addition, the MG03PS19 mutation-negative genomic signature was associated with recurrence events in non-CMS1 subjects. When we considered CRC Stage 2 and 3 combined, we identified the MG23PS19 in which recurrence events are associated with mutation-positive and mutation-negative genomic signatures in CMS1 and non-CMS1 subjects, respectively. In addition, the mutation-positive MG23PS20 genomic signature was associated with disease-free progression (Appendix A).

Next, we assessed the probability of disease-free survival in each subject associated with each genomic signature independently and in combination with the CMS subtype (Figure 3). For CRC Stage 2, subjects with the CMS1 subtype and mutation-positive (i.e., class 01, yellow line) or negative (i.e., class 02, red line) MG02PS04 genomic signature (Figure 3A, left panel) presented a significantly higher probability of disease-free survival (*p* = 0.047 and *p* = 0.0055 for class 01 and class 02, respectively) compared with non-CMS1 subjects bearing those genomic signatures. In turn, the CMS1 subtype subjects with the mutation-positive (i.e., class 01, blue line) MG02PS05 or mutation-negative (i.e., class 02, gray line) MG02PS03 genomic signatures (Figure 3A, right panel) had a significantly lower probability of disease-free survival (*p* = 0.0021 and *p* = 0.0049, respectively) than non-CMS1 subjects. For CRC Stage 3, the CMS1 subjects presented a lower probability of disease-free survival regardless of the mutational status of the MG23PS19 and MG03PS18 genomic signatures (Figure 3B), although the differences were not statistically significant (*p* = 0.42 and *p* = 0.36 for class 01 and class 02 MG03PS19 genomic signatures, and *p* = 0.45 and *p* = 0.18 for class 01 and class 02 MG03PS18 genomic signatures and, respectively). Similarly, we did not detect significant differences in the probability of disease-free survival when CRC Stage 2 and Stage 3 subjects were considered together, regardless of the CMS subtype or mutational status of the studied genomic signatures (*p* = 0.076 and *p* = 0.32 for class 01 and class 02 MG23PS19 genomic signatures, and *p* = 0.47 and *p* = 0.32 for class 02 of MG23PS20 and MG23PS19 genomic signatures, respectively) (Figure 3C).

### 2.4. Analysis of Prognostic Capability by Sequential Combination of Genomic Signatures

Next, we constructed three distinct scenarios to examine the sequential combination of these identified genomic signatures to assess their prognostic capability. Figure 4 depicts the ROC–AUC analysis using disease-free survival-based-predictions according to the CRC stage for each of these combinations. We defined three scenarios in CRC Stage 2 (Figure 4A) and Stage 3 (Figure 4B) based on the presence of the following distinctive genomic signatures: (i) mGS02R3-class02 and mGS02R5-class01 (left panel); (ii) mGS02R4-class01, mGS02R4-class02, and mGS02R5-class01 (middle panel); and (iii) mGS02R3-class02, mGS02R4-class02, and mGS02R5-class01 (right panel). The three scenarios provided high discriminatory power between CRC Stage 2 subjects who were disease-free over 60 months and those who showed recurrence (AUC > 0.90, Figure 4A). Noteworthy, scenarios (i) and (iii) showed an identical accuracy in their predictive capacity, denoting an underlying similarity amongst the two different combination sequences, most likely attributed to the genes that the genomic signatures share (Appendix A). However, the discriminatory power was more moderate in CRC Stage 3 subjects (AUC < 0.90, Figure 4B). When considering CRC Stage 2 and Stage 3 combined, we defined three scenarios different to those defined for individual stages (Figure 4C): (i) mGS23R19-class01, mGS23R19-class02, and mGS23R20-class02 (left panel); (ii) mGS23R18-class02 and mGS23R19-class01 (middle panel); and (iii) mGS23R19-class02 and mGS23R20-class01 (right panel). These scenarios also provided a more moderate discriminatory power than that observed in CRC Stage 2 individually (AUC < 0.90, Figure 4C).

### 2.5. Diagnostic Framework for Recurrence in CRC Stage 2 Using F1CDX Genomic Signatures

We developed a hypothetical diagnostic framework for clinical application in which we combined the three distinct genomic signatures identified as the top performers in CRC Stage 2 subjects (Figure 5A). The CRC stage and CMS1 subtype are determined by standard procedures, whereas three genomic signatures (i.e., MG02PS03, MG02PS04 and MG02PS05) can be assessed using the F1CDx assay. The comparison of the marker composition of these genomic signatures revealed a high overlap between MG02PS03 and MG02PS04, being both ‘Mutation-Negative’ (Figure 2B), with 32 shared markers between them (Appendix A and Figure 5B). Hence, we reasoned that the shared markers in the MG02PS03 and MG02PS04 genomic signatures constitute a potential novel marker panel for tumor recurrence, with a reduction in the number of markers in the panel at minimum. To evaluate the value of this novel 32-marker panel, we validated its performance using an independent sample set (i.e., AACR GENIE BPC CRC 2.0-PUBLIC [22]). ROC–AUC analysis showed that the 32-marker panel had high power to predict tumor recurrence in CRC Stage 2 subjects, the empirical, binormal, and nonparametric testing metrics for the ROC curves registered AUCs resulting in: (a.) 0.9524; (b.) 0.9524; and (c.) 0.9411; respectively (Figure 5C); there were also Positive Predictive and Negative Predictive Values (PPV and NPV, respectively) of PPV = 0.9737 and NPV = 0.9231. To gain a better understanding of the molecular crosstalk represented in the 32-marker panel, we assessed the variable importance of each contributing gene. We conducted a competing risks analysis for disease-free survival using the CRC Stage 2 patients included in the AACR GENIE BPC CRC 2.0-PUBLIC dataset (Figure 5D). Out of the 32 markers within this genomic signature, 25 were provided with full clinicopathological information. For each subject, 2500 random forest survival trees were generated. Subjects with progression-free survival events demonstrated higher error rates than those exhibiting outcomes consistent with disease progression events. Of note, the lower error rates in the disease progression events are associated with the high PPV value registered for the 32-marker panel.

## 3. Discussion

We have identified genomic signatures based on F1CDx to stratify the subgroups of CRC patient populations. Our findings demonstrate the presence of diagnostic indications on the genomic level that could address the lack of clinical presentations in the less aggressive stages of CRC (i.e., Stage 2) that are more apparent in later stages, such as 3 or 4. Hence, repositioning the F1CDx companion diagnostic to support clinical treatment decisions is feasible, in the context of inferring the probabilities of recurrence and concerning indications for the scheduling of chemotherapeutics. The accuracy of the genomic signatures ranges across CRC Stage 2, CRC Stage 3, and CRC Stages 2 and 3. Of note, we detected the highest accuracy for the genomic signatures of CRC Stage 2, where clinicopathological presentation does not suffice for treatment selection. Thus, genomic signatures provide a usable tool for this clinical need by inferring recurrence with high precision.

Genomic signatures demonstrate the genomic differences between Stage 2 and Stage 3 in CRC and bear translational value in discerning the implementation of different treatment approaches. This is analogous to previously published studies whereby recurrence rates are investigated using circulating tumor DNA (ctDNA) primarily for CRC Stage 3 [13]. Our work has focused primarily upon extending the diagnostic indications of the F1CDx to profile the presence and/or absence of specific mutation patterns in CRC Stage 2 rather than Stage 3. While other studies focus on more aggressive components of cancer [7,23], our research addresses the clinical inadequacies and ambiguities associated with the treatment of less advanced forms of those cancer types. A limitation of this approach is the restriction of the potential discoverability of the specific mutations associated with F1CDx as our genomic template. One of the clinical limitations that exists within this study is the sparsity of adjuvant chemotherapy treatment data, whereby the analysis of direct patient responsiveness to treatment will require further investigation. Nevertheless, we showed that the F1CDx diagnostic platform can be repositioned as a prognostic tool by identifying the previously unreported association between the mutational profile and CRC treatment responsiveness.

Importantly, we summarized the top-performing genomic signatures in a panel consisting of 32 genes identified via the intersection of genomic signatures for CRC Stage 2, results 3 and 4, and class 02. Panel summarization and combination with clinical presentation has several advantages regarding increasing the translational potential of the findings. First, it enables the identification of the most precise set of markers for a particular clinical application in a determined patient stratum. In our case, this was the prediction of recurrence in CRC Stage 2 CMS1 subtype patients. Second, it enables a transfer to another analytical platform—for example, multiplexed qPCR or targeted sequencing—in which only the relevant genomic variants are assessed, thereby reducing costs and avoiding potential regulatory and ethical conflicts when analyzing variants with no clinical relevance. Third, it significantly reduces the computational requirements for data analysis and the reporting of results, improving the turnaround time for the test in a clinical setting. Validation efforts using an independent dataset resulted in very high precision (over 95% accuracy) during the identification of CRC Stage 2 patients in which the tumor will recur. Further validation in subgroups of CRC Stage 2 patients stratified by MSS/MSI-H, tumor position and treatment type may be of interest. However, such information was not available for all the samples in the validation cohort, notably reducing the number of subjects per stratum and therefore the power of the ROC analysis. Hence, further studies targeting these populations are warranted. Of note, the recurrence rates for the CRC Stage 2 and Stage 3 patients included in this study were higher than the reported national average [24]. This observation, however, has no major impact on the results since the genomic signature represents a personalized prediction approach.

The 32-gene panel included several genes involved in cancer progression and its putative response to treatment [25,26]. Moreover, our enrichment analysis revealed findings consistent with these 32 genes that were pertinent to the changes in copy number associated with colorectal cancer development and progression [23], gene mutations important as prognostic biomarkers for colorectal cancer screening and diagnosis [26], and the upregulation of anti-apoptotic proteins that encourage epithelial–mesenchymal cellular transition (EMT, [23,25,26,27]).

## 4. Materials and Methods

A detailed description of the datasets and analytical methods is provided in the Appendix A.

*Patient population and data set*. We obtained the patient data employed in our study from the cBioPortal for Cancer Genomics using TCGA Research Network data. These data provided the clinical and molecular attributes detailed in Appendix A. Appendix A outlines the patients’ baseline characteristics by CRC Stage 2. Disease-free survival averaged 29.10 months for CRC Stage 2 and 23.56 months for Stage 3. The F1CDx assay included cancer-related alterations across 324 genes in solid tumor DNA (Appendix A). The molecular profiles of 321/324 F1CDx target genes (Appendix A) for 378 CRC cases were generated using GISTIC 2.0 (Genomic Identification of Significant Targets in Cancer, version 2.0.22) software [28].

*Data Preprocessing and Implementation*. The data preprocessing utilized the *tidyverse* R-package (version 1.3.2). For proper XAI algorithm function, we applied ‘One-Hot Encoding’ to limit the variable levels. The *mice* R-package (version 3.15.0) imputed missing values for variables with ≤40% missing data or data tied to survival aspects. The XAI algorithm generated the ‘J-index’ attribute, quantitatively scoring the subgroup populations critical to DFS outcomes. The J-Index mathematically defines each subgroup’s composition. The results with the highest J-value were identified as the primary population subgroup and each population subgroup yielded two classes. Classes 01 and 02 were denoted as ‘Mutation-Positive’ and ‘Mutation-Negative’, respectively. ‘Mutation-Positive’ was defined as ‘present’ if a mutation exists in all the genes within that class 01 genomic signature. ‘Mutation-Negative’, or the class 02 genomic signature, was defined by the ‘absence’ of any mutation of all the genes that define it. Following data preprocessing, multiple hypothesis testing was performed using the Benjamini–Hochberg procedure with a false discovery rate of Q ≤ 10%. The genomic signatures for each subgroup population were identified by matching the observed patterns to specific clinical variables and values. The genomic signatures were sorted by ascending *p*-value and collected. The *sqldf* R-package (version 0.4.11) assessed the presence or absence of mutations within a genomic signature, functioning as a unit in patient subgroups.

*Disease-Free Survival Analysis*. Kaplan–Meier survival analysis was performed using the corresponding feature vectors for both the clinical indications and identified genomic signatures from the top subgroup collections to assess the “disease-free survival” (DFS), commonly referred to as recurrence, using the R-packages *survival* (version 3.5.0), *survminer* (version 0.4.9), and *ggsurvfit* (version 0.2.1). Survival probabilities were estimated using DFS as the event measured, and a *p*-value ≤ 0.05 was considered significant.

*ROC-AUC Analytical Validation*. The identified genomic signatures were assessed for their performance using Receiver Operating Characteristic (ROC) analysis, conducted with the R-package *ROCit* (version 2.1.1) in the R environment (version 4.2.2). We calculated the empirical, binormal, and non-parametric test functions for each ROC analysis to determine which genomic signature(s) was most able to determine recurrence. To ensure a fair comparison, we reported Area Under the Curve (AUC) values for all three ROC functions, minimizing bias. We evaluated the likelihood of recurrence among patients using the top-performing genomic signatures in the training cohort (i.e., TCGA cohort) (Figure 2). In addition, we validated the 32-gene marker panel in the validation cohort (i.e., AACR GENIE BPC CRC 2.0-PUBLIC; http://cbioportal.org/genie/ (accessed on 16 February 2024)) (Figure 5C).

*Marker panel summarization and validation using independent cohort.* CRC Stage 2 subjects were compared according to the gene composition of their top three performing genomic signatures (i.e., MG02PS03, MG02PS04, and MG02PS05) using UpSet plots, implemented with the R-package *VennDetail* (version 1.14.0). Additionally, we validated a marker panel consisting of 32 genes in an independent dataset (AACR GENIE BPC CRC 2.0-PUBLIC) (Appendix A). The validation included ROC–AUC analysis, following the same methodology outlined above.

*Enrichment Analysis.* R-package *randomForestSRC* (version 3.2.3) selection was predicated on its survival forest algorithm. The threshold of ntree is 2500 for 29/32 prognostic genes when used to assess the competing risks and variable importance of the downstream analysis of the validation dataset (AACR GENIE BPC CRC 2.0-PUBLIC) [22].

## 5. Conclusions

Using a combination of next-generation sequencing and ultramodern computational techniques, we have demonstrated the translatability of the FDA-approved F1CDx companion diagnostic tool used to examine patients according to both their phenotype and genomic signatures to assess the risk of recurrence in CRC Stage 2 patients with over 95% precision. Future studies are warranted to formally assess the performance of these genomic signatures in larger populations according to regulatory guidelines.

## Figures and Tables

**Figure 1 ijms-25-03220-f001:**
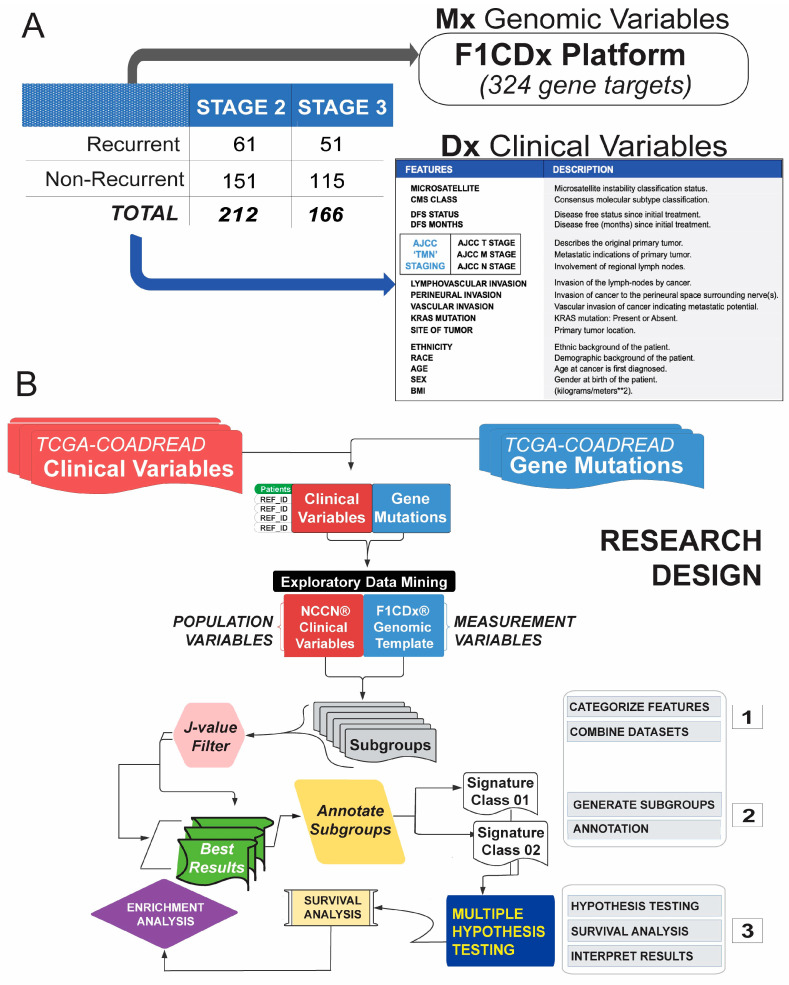
Dataset overview and experimental design. (**A**) Training data set consisted of mutational and clinical data from *n* = 378 subjects with recurrent and non-recurrent Stage 2 or 3 CRC retrieved from TCGA for which genomic variables and clinical variables were available. (**B**) Mutational and clinical features were analyzed using the XAI algorithm [18]. Subgroups were scored according to the pattern definition (i.e., J-score), and genomic signatures identifying each subgroup were generated. The performance of such genomic signatures was assessed via disease-free survival analysis, biological interpretated via gene enrichment analysis and further validated using an independent dataset.

**Figure 2 ijms-25-03220-f002:**
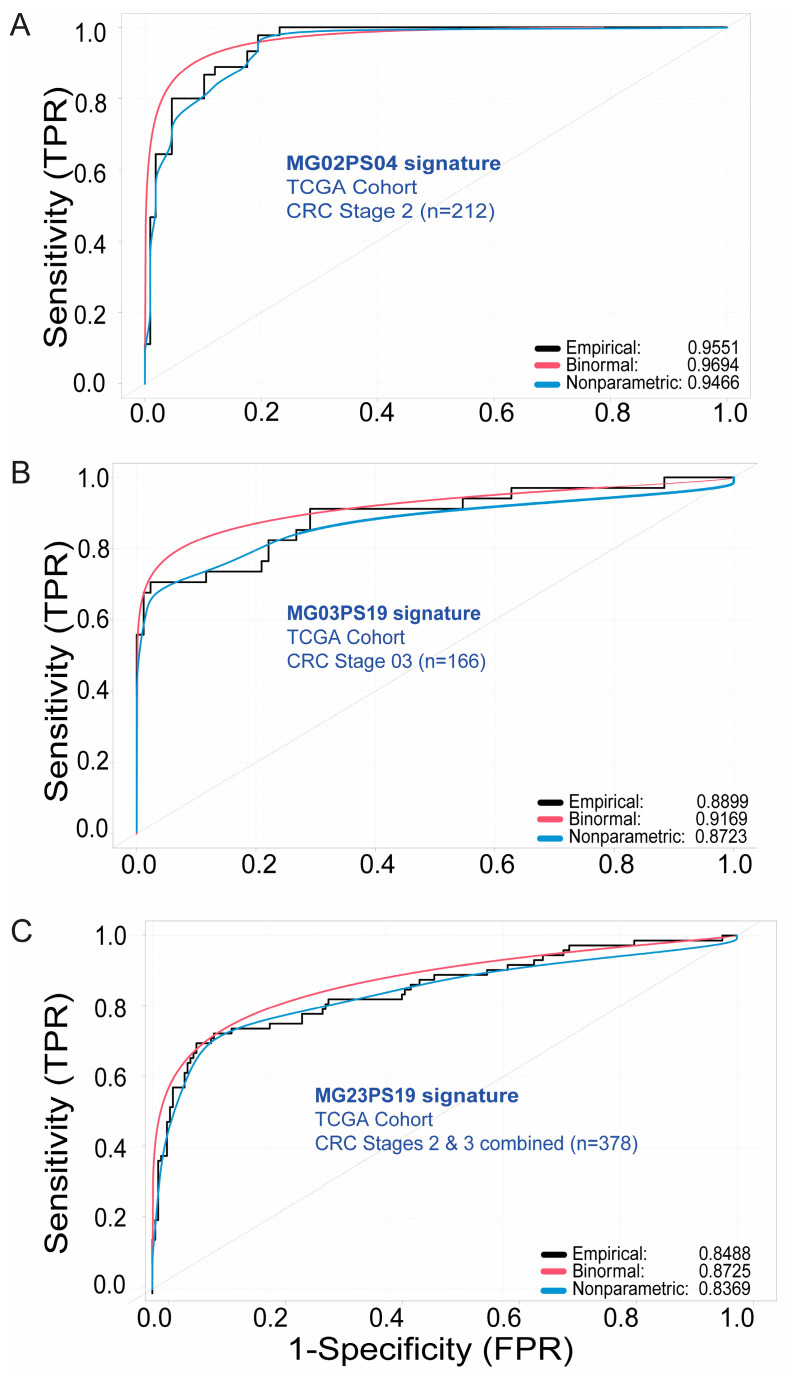
ROC–AUC analysis of genomic signatures. Performance assessment of the top genomic signatures for each CRC group and their ability to infer a recurrence event in samples from the training cohort (i.e., TCGA cohort). (**A**) Genomic signature MG02PS04 for CRC Stage 2 patients; (**B**) Genomic signature MG03PS19 for CRC Stage 3 patients. (**C**) Genomic signature MG23PS19 for Stage 2 and Stage 3 CRC patients combined. ROC: Receiver Operating Curve. AUC: Area Under the Curve. x- and y-axes represent the values for Specificity (False Positive Rate: FPR) and Sensitivity (True Positive Rate: TPR) when distinguishing between recurrent and disease-free patients.

**Figure 3 ijms-25-03220-f003:**
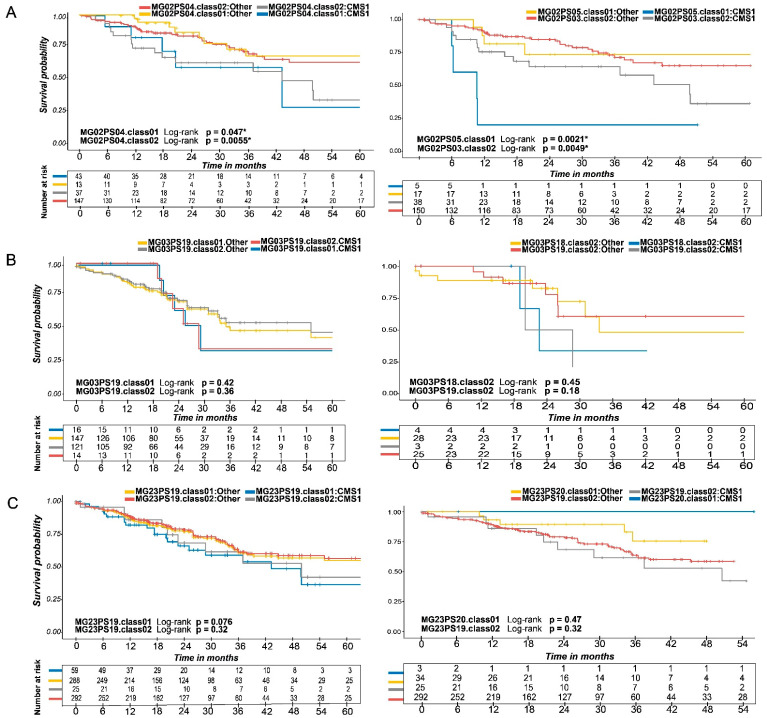
Disease-free survival analysis of groups by CRC stage. Disease-free survival given the presence and/or absence of these genomic signatures in conjunction with the clinical indications of consensus molecular subtypes. (**A**) CRC Stage 2 genomic signatures produced the only statistically significant results, with results 3, 4, and 5 retaining prognostic potential. (**Left**) Kaplan–Meier survival analysis of MG02PS04.class01 (Log-rank; *p* = 0.047) and MG02PS04.class02 (Log-rank; *p* = 0.0055). (**Right**) Kaplan–Meier survival analysis of MG02PS05.class01 (Log-rank *p* = 0.0021) and MG02PS03.class02 (Log-rank *p* = 0.0049). The purpose of the asterisks (*) was to emphasize the statistical significance of these results in comparison to the other observed outcomes. (**B**) Disease-free survival of CRC Stage 3 and the corresponding genomic signatures did not have statistically significant results. (**Left**) Kaplan–Meier survival analysis of MG03PS19.class01 (Log-rank; *p* = 0.42) and MG03PS19.class02 (Log-rank; *p* = 0.36). (**Right**) Kaplan–Meier survival analysis of MG03PS18.class02 (Log-rank; *p* = 0.45) and MG03PS19.class02 (Log-rank; *p* = 0.18). (**C**) CRC Stages 2 and 3 demonstrated some promising results; however, they lack statistical significance and/or the stratification of the identified patient populations were exceedingly small. (**Left**) Kaplan–Meier survival analysis of MG23PS19.class01 (Log-rank; *p* = 0.076) and MG23PS19.class02 (Log-rank; *p* = 0.32). (**Right**) Kaplan–Meier survival analysis of MG23PS20.class01 (Log-rank; *p* = 0.47) and MG23PS19.class02 (Log-rank; *p* = 0.32).

**Figure 4 ijms-25-03220-f004:**
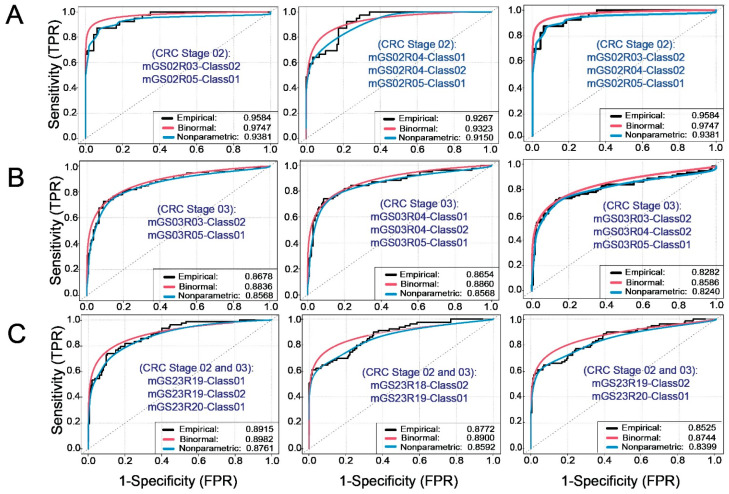
ROC–AUC analysis using disease-free survival-based-predictions according to CRC stage. The sequential combination of genomic signatures resulted in different scenarios for the enhanced predictive capability in CRC Stage 2 and 3 subjects considered separately or in combination. (**A**) ROC–AUC analysis results for the CRC Stage 2 scenarios, for empirical, binormal, and nonparametric ROCs, respectively: Scenario (i) (**left**) AUCs = 0.9584, 0.9747, 0.9381; scenario (ii) (**middle**) AUCs = 0.9267, 0.9323, 0.9150; and scenario (iii) (**right**) AUCs = 0.9584, 0.9747, 0.9381. (**B**) ROC–AUC analysis results for the CRC Stage 3 scenarios: scenario (i) (**left**) AUCs = 0.8678, 0.8836, 0.8568; scenario (ii) (**middle**) AUCs = 0.8654, 0.8860, 0.8568; and scenario (iii) (**right**) AUCs = 0.8282, 0.8586, 0.8240. (**C**) ROC–AUC analysis results of the CRC Stages 2 and 3 scenarios: scenario (i) (**left**) AUCs = 0.8915, 0.8982, 0.8761; scenario (ii) (**middle**) AUCs = 0.8772, 0.8900, 0.8592; and scenario (iii) (**right**) AUCs = 0.8525, 0.8744, 0.8399. The combination of genomic signatures for each scenario is provided in the ROC and described in the text. Empirical, binormal, and nonparametric ROCs are represented by black, red, and blue lines, respectively.

**Figure 5 ijms-25-03220-f005:**
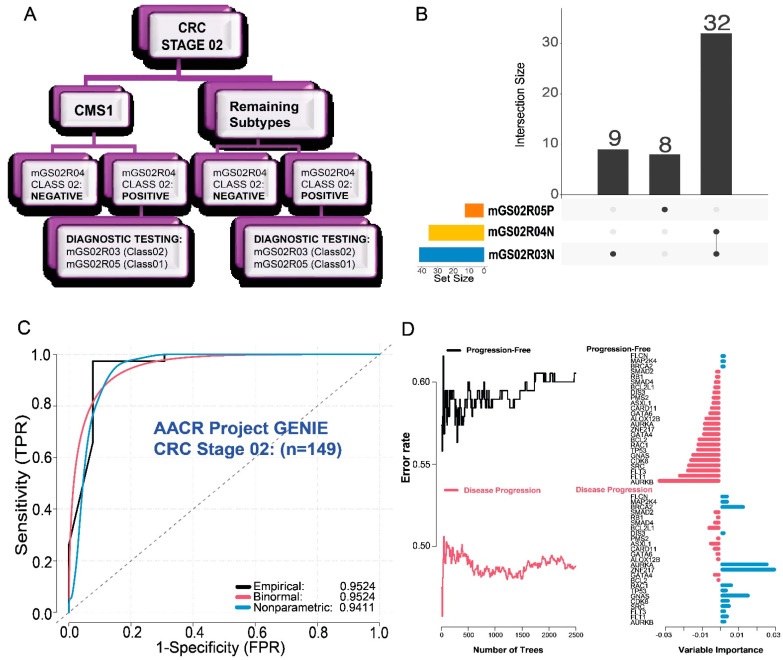
Building and validating a diagnostic framework for tumor recurrence in CRC Stage 2 patients. The summarization of the top-performing genomic signatures resulted in a 32-gene genomic signature that could predict tumor recurrence in CRC Stage 2 patients with high precision. (**A**) Hypothetical diagnostic framework created by combining standard assessment (i.e., tumor staging and CMS subtype) with the three top-performing genomic signatures in this study (MG02PS03, MG02PS04 and MG02PS05). (**B**) Overlap analysis between the top-performing genomic signatures showed that 32 markers were shared between the MG02PS03 and MG02PS04 genomic signatures representing a potential novel marker panel. (**C**) Validation of the 32-marker panel in an independent dataset. ROC–AUC analysis showed high power to predict tumor recurrence in CRC Stage 2 subjects, with AUCs = 0.9524, 0.9524, and 0.9411 for empirical (black line), binormal (red line), and nonparametric (blue line). (**D**) Random survival forests were implemented as a representation of the 32-marker panel using variable importance for AACR GENIE BPC CRC 2.0-PUBLIC according to the outcomes for patients with either ‘Disease Progression’ (**bottom**) or those classified as ‘Progression-Free’ (**top**). Left panel: y-axis represents the error rates for each iteration of survival tree generated, while the x-axis represents the number of trees. Right panel: Variable importance is represented in the x-axis. Positive (blue bars) and negative (red bars) represent upregulated and downregulated genes in CRC, respectively.

## Data Availability

CRC patient data for the training cohort were accessed using the cBioPortal for Cancer Genomics (http://cbioportal.org/ (accessed on 15 June 2023)) to collect the appropriate molecular and clinical attributes for our study based upon data generated by the TCGA Research Network under the Broad Institute GDAC TCGA Analysis Pipeline License. The clinical attributes used to generate our collection of observed population subgroups are in Appendix A. The molecular profiles of 321/324 of the F1CDx target genes (Appendix A), extracted as copy–number alterations (CNAs) and represented as putative copy–number calls for 378 cases using GISTIC 2.0 (Genomic Identification of Significant Targets in Cancer, version 2.0.22), are available from URL: (https://doi.org/10.1186/gb-2011-12-4-r41 (accessed on 15 June 2023)). AACR GENIE BPC CRC 2.0-PUBLIC (http://cbioportal.org/genie/ (accessed on 15 June 2023)) was used for validation, whereby 29/32 of the genes identified were available for analysis.

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
