# Peer review of "Identification of Genomic Signatures for Colorectal Cancer Survival Using Exploratory Data Mining"

_ijms, 2024, doi:10.3390/ijms25063220_

Round 1

Reviewer 1 Report

Comments and Suggestions for Authors

The article provides a thorough analysis of colorectal cancer (CRC) and discusses the potential use of the FoundationOne Companion Diagnostic (F1CDx) for recurrence prediction in stage 2 patients. This is achieved through the use of a novel exploratory algorithm based on artificial intelligence (XAI), which identifies multi-gene prognostic signatures by integrating clinical and mutational features. There are a few inconsistent aspects, though, and some recommendations for enhancement.

The phrase "XAI algorithm" is introduced in the paper, although readers are not given a precise description or explanation. When mentioning the XAI algorithm for the first time, it is important to provide readers a quick explanation or definition so they may comprehend its importance and how it relates to the study.

To improve repeatability and openness, include more information in the text regarding the data's source, such as the TCGA (The Cancer Genome Atlas) dataset.

The statement, "The recurrence rates observed in CRC stage 2 and stage 3 patients exhibit a higher than the national average reports," appears to be unclear. For improved readability, think about editing. Throughout the article, the phrases "CRC stage 2" and "CRC Stage 2" are interchangeable. For clarity, it's critical to keep capitalization consistent.

For easier reading, the complicated line "By utilizing our novel XAI algorithm to assess genomic mutations based on the F1CDx test, we built prognostic signatures to determine risk probabilities of recurrence events in CRC stage 2 patients" may be rewritten.

In the introduction, on line 56, the phrase "prognostic determinations" would be better understood as "prognostic predictions" or "prognostic assessments."

Comments on the Quality of English Language

Moderate editing of English language required

Author Response

The article provides a thorough analysis of colorectal cancer (CRC) and discusses the potential use of the FoundationOne Companion Diagnostic (F1CDx) for recurrence prediction in stage 2 patients. This is achieved through the use of a novel exploratory algorithm based on artificial intelligence (XAI), which identifies multi-gene prognostic signatures by integrating clinical and mutational features. There are a few inconsistent aspects, though, and some recommendations for enhancement.

The phrase "XAI algorithm" is introduced in the paper, although readers are not given a precise description or explanation. When mentioning the XAI algorithm for the first time, it is important to provide readers a quick explanation or definition so they may comprehend its importance and how it relates to the study.

Definition of XAI algorithm (i.e., explainable artificial intelligence) is now included in the abstract and when mentioned in the text for the first time. In addition, a brief explanation of the concept is now added in the last paragraph of the introduction, complementing the existing explanation of the XAI algorithm (page 2, lanes 67-79)

To improve repeatability and openness, include more information in the text regarding the data's source, such as the TCGA (The Cancer Genome Atlas) dataset.

Details on the TCGA dataset are included in the Supplementary Material file under “Detailed Material and Methods”.

The statement, "The recurrence rates observed in CRC stage 2 and stage 3 patients exhibit a higher than the national average reports," appears to be unclear. For improved readability, think about editing. Throughout the article, the phrases "CRC stage 2" and "CRC Stage 2" are interchangeable. For clarity, it's critical to keep capitalization consistent.

The statement was moved to the Discussion section (page 12, lines 390-393) and edited accordingly. Furthermore, we used the capitalized form for “Stage” throughout the manuscript.

For easier reading, the complicated line "By utilizing our novel XAI algorithm to assess genomic mutations based on the F1CDx test, we built prognostic signatures to determine risk probabilities of recurrence events in CRC stage 2 patients" may be rewritten.

We reworded this sentence as it follows: “To this end, we applied our novel XAI algorithm to assess F1CDx test results to build a molecular signature for determining recurrence risk in CRC stage 2 patients.” (Page 2, lines 77-79)

 In the introduction, on line 56, the phrase "prognostic determinations" would be better understood as "prognostic predictions" or "prognostic assessments."

We changed the term to “prognostic assessments”.

Reviewer 2 Report

Comments and Suggestions for Authors

The authors used their developed XAI algorithm to identify genomic signatures for CRC. It was a clinical application research, which requires precision. However, the manuscript was full of errors, short of precision, and hard to understand.

1)      TCGA was the data source the authors used. However, TGCA, instead of TCGA, is all over the manuscript, including the abstract, which is no good at all.

2)      The paragraph (Lines 89-98) is difficult to understand, with data in supplementary tables S3 and S4 not matching with each other, classes 01 and 02 not explained (only found in MM section), and primary population not numbered. When prognostic signature was defined by gene mutations, the authors added “absent of all mutational elements”, which was truly confusing. When I found data in Table S5 were close to those in S4, the counts were not matched exactly, which was deeply troubling.

3)      Figure 2 is hard to read too. ROC-AUC, TPR, or FPR was not full-named, signature was not defined (Class 01 genes cannot be found), population was not numbered, mutation-positive or -negative was not defined well. Was any mutation (sense and non-sense) in a gene counted? Were the figures generated from 32-gene marker panel? Any justification?

4)      Following MG02PS04, MG03PS19, and MG23PS19, all other signatures were not defined, including MG02PS03, MG02PS05, MG03PS18, MG23PS20, etc. The authors used signature, genomic signature, prognostic signature, and molecular signature, terms of which are hard to distinguish without interpretation.

5)      There were no Figure 1C or 1D (Lines 343-344) in Figure 1.

Comments on the Quality of English Language

Some sentences are too long to read.

Author Response

The authors used their developed XAI algorithm to identify genomic signatures for CRC. It was a clinical application research, which requires precision. However, the manuscript was full of errors, short of precision, and hard to understand.

We thank the reviewer for the insightful review of our manuscript. Errors have been corrected in this version of the manuscript.

1)      TCGA was the data source the authors used. However, TGCA, instead of TCGA, is all over the manuscript, including the abstract, which is no good at all.

We reviewed and corrected the use the appropriate acronym (i.e. TCGA) throughout the whole manuscript.

2)      The paragraph (Lines 89-98) is difficult to understand, with data in supplementary tables S3 and S4 not matching with each other, classes 01 and 02 not explained (only found in MM section), and primary population not numbered. When prognostic signature was defined by gene mutations, the authors added “absent of all mutational elements”, which was truly confusing. When I found data in Table S5 were close to those in S4, the counts were not matched exactly, which was deeply troubling.

The paragraph was completely rewritten for clarity, including an explanation on the class definition and the type of mutation (Pages 4-5, lines 131-140). We also provide a revised Supplementary Table S4 in which the number of genes corresponds to those listed in Supplementary Table S3.

3)      Figure 2 is hard to read too. ROC-AUC, TPR, or FPR was not full-named, signature was not defined (Class 01 genes cannot be found), population was not numbered, mutation-positive or -negative was not defined well. Was any mutation (sense and non-sense) in a gene counted? Were the figures generated from 32-gene marker panel? Any justification?

Figure 2 presents the initial performance assessment for each genomic signature individually to distinguish between recurrent and disease-free patients. Studied signatures were those detailed in Supplementary Tables S3 and S4. Any mutation (sense or non-sense) in a gene was counted. The 32-marker genomic signature was built by combining the top performers genomic signatures and validated in an independent cohort (Please refer to Figure 5 in the manuscript)

For clarity, in this revised version of the manuscript, Figure 2 and its legend were modified to include the name and number of patients for each signature. All acronyms’ definitions were included in the legend, as well as the main text. Furthermore, the paragraph in the text was completely rewritten for clarity (Page 5 lines 157-171).

4)      Following MG02PS04, MG03PS19, and MG23PS19, all other signatures were not defined, including MG02PS03, MG02PS05, MG03PS18, MG23PS20, etc. The authors used signature, genomic signature, prognostic signature, and molecular signature, terms of which are hard to distinguish without interpretation.

The composition of all signatures cited on the manuscript are now included in the edited version of the Supplementary Table S4. We also edited the Supplementary Table S5 for clarity and correct counting of the genes in Signature MG02PS03 – Class 02.

For clarity we now use the term “genomic signature” throughout the manuscript.

5)      There were no Figure 1C or 1D (Lines 343-344) in Figure 1.

The reference for Figures 1C and 1D in the main text were erroneously kept from a previous version of the manuscript. The corresponding paragraph in the Materials and Methods section has been corrected (Page 12, lines 393-402) and included references to the correct figures.

Reviewer 3 Report

Comments and Suggestions for Authors

Manuscriot entitled "Identification of genomic signatures for colorectal cancer survival using an exploratory data mining."

Since there are too many signatures predicting survival, the authors should provide solid validation and novelty to make this work publishable.

1. The author should validate the signature in MSS, MSI-H, populations, left side and right side colon.

2. The authors should stratify patients with various treatment protocol.

3. The baseline data of their cohorts toether with main mutations should be summarized and presented.

Comments on the Quality of English Language

Aceptable

Author Response

Manuscriot entitled "Identification of genomic signatures for colorectal cancer survival using an exploratory data mining."

Since there are too many signatures predicting survival, the authors should provide solid validation and novelty to make this work publishable.

  1. The author should validate the signature in MSS, MSI-H, populations, left side and right side colon.
  2. The authors should stratify patients with various treatment protocol.
  3. The baseline data of their cohorts toether with main mutations should be summarized and presented.

Baseline patient characteristics are provided in Supplementary Table 2. Validation of the 32-marker genomic signature has been conducted using an independent dataset (i.e., AACR Project GENIE) resulting in ~95 % accuracy (AUCs= 0.9524, 0.9524, and 0.9411 for empirical, binormal, and nonparametric ROCs, respectively) to predict tumor recurrence in CRC Stage 2 patients. These results are presented in Figure 5 of the manuscript. Further stratification according to MSS/MSI-H, tumor position and treatment type was hindered by the lack of such information for all samples in the database, reducing notably the number of subjects per stratum and therefore, the power of the ROC analysis. We acknowledge this point in the revised version of the manuscript (Page 11, lines 336-344)

Notwithstanding, upon the reviewer’s request, we conducted ROC analyses evaluating the 32-marker genomic signature in samples from the validation cohort stratified according to MSS/MSI-H, tumor position. and treatment type, when available (see figure in attachment). In all cases, we observed a remarkable reduction in accuracy supporting the need for an increased sample size per stratum.

Figure 1: Performance of the 32-marker genomic signature in the validation cohort stratified according to MSS/MSI-H, tumor position and treatment type. The performance of the 32-markers genomic signature in distinguishing patients with recurrent tumors from those with disease-free progression was evaluated in subgroups of the validation cohort stratified by MSS/MSI-H status (Panel A), tumor position (Panel B), and treatment type (Panel C) by Receiving Operating Curve (ROC) analyses. Combined treatment: Patients receiving Fluorouracil, Leucovorin Calcium, Oxaliplatin. Single-drug treatment: Patients receiving only Capecitabine.

Round 2

Reviewer 1 Report

Comments and Suggestions for Authors

The authors have implemented all the requested modifications, and the text has been revised accordingly. No additional chamges are necessary.

Reviewer 2 Report

Comments and Suggestions for Authors

Accept in present form.

Reviewer 3 Report

Comments and Suggestions for Authors

The revision is acceptable in the present form.

Comments on the Quality of English Language

Acceptable